# An Explainable Machine Learning Framework to Inform Integrative Psychosocial Correlates of Sleep Functioning in Adults with Cancer and their Caregivers

Jerry Bonnell[1], Nikhita Guhan[2], Thomas C. Tsai[3], Robert Moulder[4], Mitsunori Ogihara[2], Youngmee Kim[3]

*Abstract*—Sleep disturbance in cancer patients and caregivers is a substantial challenge in survivorship care. As a dyadic process influenced by daytime experiences, understanding interdependent sleep health by identifying critical dyadic stress regulatory factors and psychosocial predictors is crucial for informing effective interventions. This supervised machine learning (ML) study utilized multi-modal, multi-level data from patients with colorectal cancer and spousal caregivers (n = 149 dyads; 298 persons). The dataset integrated psychosocial characteristics, dyadic cardiovascular and psychological responses to laboratory-induced stress, and 20 self-report and actigraph-derived sleep markers. Preprocessing consisted of three optional techniques: principal component analysis (P) for high dimensionality, correlated feature selection (C) for multicollinearity, and data augmentation (A) for small sample size. Four regression algorithms (Linear Regression, Ridge, Random Forest, and Support Vector Regression) were trained independently for each sleep outcome, evaluating optimal performance across different preprocessing combinations. SHAP analysis was subsequently utilized on best-fitted models to identify key predictors. Linear Regression best predicted caregivers' actigraph-derived interdaily stability ($R^2$=31.2%, via P+C+A), while Ridge best predicted patients' self-reported sleep efficiency ($R^2$ = 18.0%, via P+A), improving non-preprocessed baselines by 30.4% and 11.7%, respectively. SHAP identified dyadic stress regulatory indices as key predictors with complex dynamics and subsequent commonality analysis revealed phase-specific suppressor effects among such indices. This study identified optimal preprocessing combinations that significantly improved sleep prediction in oncology dyads. Complex interactions observed among key dyadic stress regulation indices map out pathways of stress response that shape sleep health, advancing understanding of its interdependent nature and informing targeted intervention strategies.

## I. INTRODUCTION

The growing U.S. cancer survivor population, projected to increase from 16.9 million to 26.1 million by 2040 [1], poses significant public health challenges. This population faces distinct vulnerabilities, exhibiting significantly higher morbidity and mortality rates than the general U.S. population [2]–[7]. Sleep disturbance – defined by difficulty falling asleep and frequent, prolonged nighttime awakenings – is highly prevalent in cancer survivors (33-59%, as opposed to 15-20% in the general population) and their family caregivers (36-95%) [8]. Such disturbance is further associated with poor quality of life, circadian dysregulation, other major diseases, poor cancer prognosis, recurrence, and mortality in cancer survivors, and with degraded caregiving quality, increased morbidity risk, and reduced quality of life in their caregivers [8].

Since sleep is often affected by various daytime stress experiences not only at the individual level but also the dyadic level, identifying critical factors associated with respective stress regulatory patterns would inform effective interventions and improve health outcomes for this already vulnerable population. However, traditional unidimensional approaches are insufficient to fully capture the complexity of multi-dimensional and multi-level factors involved in sleep health of both patients and caregivers simultaneously [9], [10].

Machine learning (ML) is particularly well-suited to address these complexities. Specifically, supervised ML, including linear models, tree-based methods, and Support Vector Machines (SVM), have been applied to predict perceived sleep quality by establishing quantitative links between objective data (e.g., actigraph-derived time in bed, sleep duration, number of awakenings, and movement indices) and subjective sleep assessments (e.g., self-report sleep onset latency, wake after sleep onset, and sleep duration) [10], [11]. Studies have also included non-sleep variables, such as sociodemographic factors, physical and mental health, health behaviors, and medication use in the model as predictors [9], [12].

Such ML methods have been effective in pinpointing key correlates of sleep health. For instance, in 69 community-dwelling older adults with dementia, CatBoost identified actigraph-derived sleep irregularity and medication burden as primary predictors of poorer sleep efficiency from sleep indices, dementia-related information, functional measures, and cytokines [13]. In another study with 3,173 community-dwelling men and women aged between 39 and 90, Lasso and Random Forest retrieved polysomnography-derived sleep efficiency, WASO, along with age, as top predictors for sleep depth and restfulness from demographic, clinical, polysomnography, and quantitative EEG variables [14].

[1]Jerry Bonnell is with the Frost Institute for Data Science and Computing, University Of Miami, Coral Gables, FL 33124, USA

[2]Nikhita Guhan and Mitsunori Ogihara are with the Department of Computer Science, University Of Miami, Coral Gables, FL 33124, USA

[3]Thomas C. Tsai and Youngmee Kim are with the Department of Psychology, University Of Miami, Coral Gables, FL 33124, USA

[4]Robert Moulder is with the Institute of Cognitive Science, University of Colorado Boulder, Boulder, CO 80309, USA

To this end, Explainable AI (XAI) methods, including post-hoc explainability techniques like SHAP (SHapley Additive exPlanations), have gained traction in the literature and have helped clarify the directional contributions of specific features on sleep predictions [15], [16]. However, interpreting SHAP contributions can be non-trivial when faced with multi-level factors that are highly intercorrelated. In the case of linear models, Commonality Analysis (CA) has been used to quantify complex interactions like suppressor effects that can arise among such predictors by partitioning explained variance into unique and shared components [17], [18].

Our work introduces a novel analytical approach: the first multi-level ML framework aimed at discovering patterns of interdependence that influence sleep health within patient-caregiver dyads. While prior work has applied ML to multi-dimensional data at the individual level, our approach uniquely leveraged an intensive multi-modal database collected from such patient-caregiver pairs, integrating laboratory-induced stress responses and self-report questionnaire data to identify the most critical biopsychosocial variables involved in sleep health of adults with cancer and their caregivers. Also, this ML study examined for the first time multiple sleep health outcomes within such an interdependent dyadic context, aiming to identify key predictive factors operating across levels (see Figure 1).

More specifically, this paper contributes the following:

1) An evaluation of specific data preprocessing techniques to identify combinations that best improve out-of-sample prediction accuracy in sleep health models for oncology dyads.
2) The application of XAI methods to retrieve salient predictors of sleep health, which ranked dyadic stress regulatory indices as top factors among a set of 880 integrated multi-modal and multi-level features.
3) The use of Commonality Analysis to clarify complex interactions among multi-level features, which revealed significant phase-specific suppressor effects, thereby supporting a dyadic predictive approach for sleep health.

## II. METHODS

### A. Data Collection

*1) Procedure:* The present study is a secondary analysis that utilized initial assessment data (T1) from 149 patient-caregiver dyads ($N$ = 298 individuals) who participated in a registered longitudinal study examining associations of cancer-related stress with health outcomes [19]. The original study received Institutional Review Board approval. Patients were identified by medical records of participating clinics and screened to determine their eligibility to participate in the study. Eligible patients were asked to nominate a spousal/partner caregiver, who were then screened for eligibility. On the day of assessment, participants completed questionnaires individually and underwent a stress induction procedure together. Participants also completed daily sleep logs for 14 consecutive days while wearing an actigraph on the wrist of the non-dominant hand.

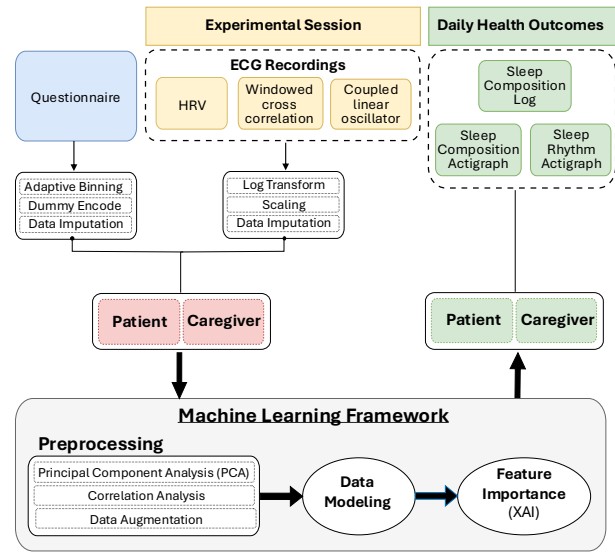

Fig. 1. Schematic representation of the supervised ML framework. The data are subjected to a preprocessing module with optional steps before data modeling. Salient predictive features are identified by feature importance analysis using XAI methods.

*2) Self-report questionnaire data:* Participants completed a self-administered questionnaire organized into two parts. The first collected socio-demographic information (age, gender, ethnicity, education, income, employment status, household composition), as well as health behaviors and indicators (smoking status, alcohol intake, body mass index, and medical conditions). The second assessed individual differences in personality, emotion regulation, stress and coping, and sociocultural factors (optimism [20], Big Five personality traits [21], self-control [22], behavioral activation/inhibition (BAS/BIS) [23], [24], perceived stress [25], coping with cancer stress [26], cancer-related stress appraisal [27], loneliness [28], perceived social support [29], social support network size [30], biculturalism [31], and familism [32]) and relationship quality with the partner (attachment quality [33], [34], and relationship satisfaction [35]–[37]). This section also included caregiver-specific measures, i.e., dimensions of care tasks [34], activities of daily living (ADL/IADL) [30], [38], and the duration, stress [39], motivation [40], and experiences [41] of caregiving.

*3) Acute stress response data:* Participants underwent the STITCH task [42], [43], which involves six phases: baseline (B), scenario presentation (S), speech preparation (P), caregiver speech delivery (SP1), patient speech delivery (SP2), and recovery (R). Multiple measurements were collected throughout the session, including continuous Heart Rate Variability (HRV) and interbeat interval (IBI) data, using a BioNex 8 Slot Chassis modified for dyadic data collection, with participants individually connected via seven unipolar leads equipped with Ag/AgCl spot electrodes. These data enabled the calculation of cardiovascular synchrony parameters using two different methods. First, stress coregulation and stress coagitation indices were derived with windowed cross correlation. Second, a coupled linear oscillator (CLO) model was used to derive

three parameters gamma, eta, and zeta, which represent the difference between partner's IBI and one's own IBI at a momentary point in time.

*4) Sleep outcomes:* Participants self-reported sleep daily upon waking using the Consensus Sleep Diary [44]. From these diaries, 14-day averages were calculated for sleep onset latency (SOL; minutes between intending to sleep and sleep onset), wake after sleep onset (WASO; minutes awake between sleep onset and final awakening), sleep duration (SD; hours asleep, derived as [time from intending to sleep to final awakening] – SOL – WASO), total time in bed (TB; hours at T1), and sleep efficiency (SE; [SD / TB] * 100).

SOL, WASO, and SD were also assessed using an actigraph. These composition indices were quantified from 60-second epochs of actigraph-derived activity counts using the Cole-Kripke algorithm, a validated method in sleep research [45]–[47]. Actigraph-detected in-bed and out-of-bed times were manually adjusted to align with self-reported times before deriving such actigraph-measured sleep indices.

Actigraphy was also used to derive sleep rhythm markers. Intradaily variation (IV; range 0-2) indicated rest-activity fragmentation and was calculated from mean squares of hourly activity differences [48]. Interdaily stability (IS; range 0-1) quantified 24-hour rhythm stability and synchronization to the light-dark cycle and was derived by normalizing the 24-hour value from a chi-square periodogram [48]. Sleep Regularity Index (SRI; range 0-100) assessed sleep-wake state consistency between corresponding times on different days, and was averaged over the 14-day period [49].

## B. Preprocessing

*1) Questionnaire:* Questionnaire response distributions exhibited skewness with certain item options endorsed infrequently. An adaptive binning procedure was implemented that grouped response categories based on their frequency of occurrence. The approach preserved commonly selected responses as individual categories while consolidating less frequent responses.[1] Following binning, questionnaire responses were dummy encoded. Missing values in this modality were treated as Missing Not At Random (MNAR), with missingness itself encoded as a distinct category. The preprocessed questionnaire data generated 850 dummy features.

*2) Acute stress & sleep indices:* All continuous sleep health outcomes were standardized, while acute stress indices were first log-transformed to address skewness and then standardized. Coregulation and coagitation indices showed near-zero correlations with sleep health outcomes while also exhibiting multicollinearity across experimental phases. We retained all such stress regulatory indices due to potential suppression effects. In contrast, predictors from the CLO model, which were also found to exhibit multicollinearity, did not demonstrate such effects and, therefore, only the gamma component from

the caregiver speech phase was retained for both patients and caregivers. Missing values in acute stress indices were imputed using column-wise means, based on the assumption that missing data resulted from participants' inability to complete the experimental protocol. Thus, the preprocessed data retained 6 HRV measures, 12 coagitation and coregulation indices, 1 gamma CLO-derived feature, and 20 sleep health indices for each patient and caregiver.

*3) Preprocessing Module:* We implemented a modular preprocessing module designed to generate and evaluate different integrated datasets resulting from various configurations. First, an option for Principal Component Analysis (PCA) was implemented as a denoising strategy to address high dimensionality and noise in questionnaire-derived features. The number of principal components was chosen to explain at least 40% of the cumulative variance, which was determined empirically by out-of-sample prediction performance, and these components were then standardized. Second, different feature selection strategies were evaluated. A key comparison involved assessing the performance impact of retaining the full predictor set (post-PCA, if applied), acknowledging potential suppressor effects, versus applying a correlation filter. When the correlation filter was applied, predictors were selected based on their Pearson correlation coefficient with a given outcome variable, retaining only those exceeding an absolute correlation threshold of 0.1. Third, the module included a mechanism to address imbalance in target variable distributions. When employed, this procedure binned a given outcome variable into five pseudo-classes and performed sampling with replacement to generate a balanced training set (i.e., 1,500 virtual samples from 300 per pseudo-class). Pseudo-classes were used only for data augmentation, not direct training targets. Following preprocessing, all patient and caregiver features were concatenated into a single input vector for model training.

## C. Data Modeling

A supervised learning testbed was developed to evaluate the predictive capacity of the integrated and preprocessed datasets, as described in Section II-B. Four primary models were selected that encompass both interpretable and flexible methods: Linear Regression (LR), Ridge Regression (RIDGE), Support Vector Regression with linear kernel (SVR), and Random Forest (RF). All models were implemented using the scikit-learn package in Python and evaluated using toolkit-suggested hyperparameter settings.

The supervised testbed systematically assessed all possible combinations of the three options in the preprocessing module: PCA (P), correlation-based filtering (C), and data augmentation (A). This resulted in 640 distinct model configurations (4 models × 8 preprocessing combinations × 20 outcome variables). To maximize utilization of the limited cohort size for training, leave-one-out cross-validation (LOOCV) was employed, with all preprocessing applied exclusively within each of the 149 training folds. To quantify uncertainty in predictions, cross-validated predictions were then bootstrapped. Resampled predictions were compared against ground truth

---

[1]The procedure is performed such that, after values are sorted in ascending order, individual values with frequencies exceeding 30 observations are preserved as distinct buckets, while consecutive values with lower frequencies are grouped until their combined frequency exceeds said threshold.

values to retrieve bootstrapped $R^2$ and Root Mean Square Error (RMSE) scores. The process was repeated 1,000 times to derive approximate 95% confidence intervals for each performance metric.

All computational experiments were conducted on a HPC cluster environment with Intel Xeon E5-2670 CPUs (2.60GHz, 16 cores total). Nodes were configured with dual 8-core processors and 64GiB RAM. The code for these experiments is made available online [19].

## III. RESULTS

### A. Sleep Health Prediction in Three Measurement Modalities

We evaluated the predictive performance of different model and preprocessing combinations across 20 sleep health outcomes. Model performance was assessed using resampled $R^2$ and RMSE scores following LOOCV, with approximate 90% and 95% confidence intervals computed for each configuration. To control for multiple testing effects, False Discovery Rate (FDR) correction was applied when determining significant results. The optimal model for each outcome was selected based on maximum cross-validated $R^2$ score among configurations that explained significant non-trivial variance. Figure 2 presents the results across all outcomes and Table I summarizes the $R^2$ and RMSE results for models achieving significance at the 95% level.

Of the 640 different model configurations tested, 15 (2.3%) achieved significance at the 95% level. Best results were mainly observed in actigraph-derived sleep rhythm indices. Specifically, for predicting caregivers' Interdaily Stability (IS_fm), LR, RIDGE, and SVR achieved 31.2%, 30.9%, and 27.1% explained variance, respectively, where P+C+A improved all three relative to baselines that did not contain any of the 3 preprocessing options. Significant results were also obtained for predicting caregivers' Intradaily Variability (IV_fm) with LR (P; 9.4%) and RF (P; 10.1%), and predicting patients' Sleep Regularity Index (SRI_pt), where RF achieved 9.9% explained variance (A). However, this last result was outperformed by a baseline RF model, which achieved 15.3%.

Prediction of actigraph-derived composition indices was comparably more difficult, with only 1 model achieving significance at the 95% level. RF significantly predicted caregivers' average WASO (avgWASO_min_fm) with 8.5% explained variance (C+A), which outperformed its baseline counterpart.

For self-report sleep indices, significant predictions were primarily achieved for patient-level measures. Patients' sleep efficiency (se36_mean_pt) was best predicted by LR (17.8%), RIDGE (18.0%), and SVR (12.1%), where all 3 models improved baselines following P+A. RF after preprocessing with A also improved prediction of average TB (tb110_mean_pt) compared to its baseline, achieving 12.7% explained variance.

The addition of data augmentation (A) consistently improved model performance across multiple outcomes. For caregivers' Interdaily Stability (IS_fm) prediction, adding this step to LR and RIDGE trained following P+C improved prediction by 6.5% and 5.7%, respectively, representing improvements of 30.4% and 30.4% over baselines. Similar gains were observed

TABLE I
BEST PERFORMING MODELS FOR PREDICTION OF SLEEP HEALTH OUTCOMES. BEST RESULTS FOR AN OUTCOME ARE SHOWN IN BOLD.

| Outcome (Modality) | Model | Preproc | $R^2$ | RMSE | Baseline $R^2$ | Baseline RMSE |
|---|---|---|---|---|---|---|
| IS_fm (actrhythm) | LR | P+C | 24.7% | **1.158** | 0.759% | 1.91e12 |
| | | P+C+A | **31.2%** | 1.352 | | |
| | RIDGE | P+C | 25.2% | 1.159 | 0.463% | 1.254 |
| | | P+C+A | 30.9% | 1.347 | | |
| | SVR | P+C+A | 27.1% | 1.366 | 0.472% | 1.223 |
| IV_fm (actrhythm) | RF | P | **10.1%** | **0.965** | 7.3% | 0.980 |
| | LR | P | 9.4% | 1.074 | 1.4% | 4.47e12 |
| SRI_pt (actrhythm) | RF | A | 9.9% | 0.999 | **15.3%** | **0.934** |
| avgWASO _min_fm (actcomp) | RF | C+A | **8.5%** | 1.217 | 1.5% | **1.087** |
| se36_mean _pt (sleep) | RIDGE | P | 13.5% | **0.996** | 6.3% | 1.099 |
| | | P+A | **18.0%** | 1.058 | | |
| | LR | P | 14.6% | 1.032 | 5.0% | 1.21e12 |
| | | P+A | 17.8% | 1.072 | | |
| | SVR | P+A | 12.1% | 1.234 | 5.9% | 1.095 |
| tb110_mean _pt(sleep) | RF | A | **12.7%** | **0.957** | 2.7% | 1.010 |

for patients' SE prediction (se36_mean_pt), where this step improved LR and RIDGE models trained after P by 3.2% and 4.5%, corresponding to 12.8% and 11.7% improvements over baselines. The RF model predicting patients' TB prediction (tb110_mean_pt) also benefited, showing a 10% improvement over its baseline.

The benefit brought by C-based filtering was dependent on outcome type. This step consistently improved prediction of actigraph-derived indices (actrhythm, actcomp) but saw no improvement when applied to models predicting self-report sleep log indices. The divergence is likely reflective of the impact of removing potential suppressor variables, which can be critical to predictive performance. For actigraph-based measures, the multicollinearity reduction achieved by this step outweighed any negative impact from removing suppressors.

Incorporating principal components (P) derived from questionnaire features improved performance in 12 of the 15 significant models. Of the three remaining, all are RF models, which suggests that the non-parametric approaches tested here may benefit more by learning directly from the questionnaire features.

### B. Feature Importance Analysis of Experimental-based Predictors

To identify predictors salient to sleep health outcomes, we conducted a SHAP feature importance analysis in our highest-performing models, as described in Section III-A. We aimed to identify a set of predictors that showed significance across different outcomes and establish their specific associations. Given that questionnaire features were transformed into principal components that are less directly interpretable, SHAP results were filtered to focus only on predictors derived from the experimental session.

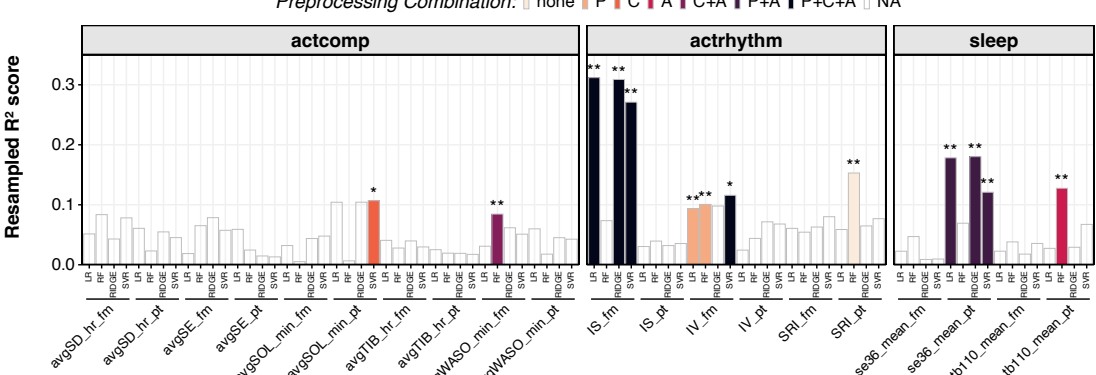

Fig. 2. Resampled mean $R^2$ scores for models predicting sleep health indices across the three modalities: actigraph-derived composition (actcomp) and rhythm (actrhythm), and self-report sleep logs (sleep). Outcomes are suffixed by participant type: patients (pt) and caregivers (fm). Single asterisks (*) indicate significant model predictions at the 90% confidence level ($p < 0.1$) and double asterisks (**) denote significance at the 95% level ($p < 0.05$). Bars shown without color (NA) indicate significant predictions were not obtained for any combination of the preprocessing steps.

We selected 11 of the 640 supervised models by first isolating those significant at the 95% level and then choosing the most accurate configuration for each model type. To ensure consistent interpretation across different model types and to account for variability in LOOCV, this analysis was based on SHAP values aggregated across folds. Predictors were extracted that (1) ranked in the top half of all unique predictors identified during the model's LOOCV evaluation, based on its aggregated mean absolute SHAP value, and (2) whose values showed a significant correlation ($p < .10$) with their corresponding SHAP values. These are visualized in Figure 3, where predictors are stratified by experimental phase and participant type (patient/caregiver).

Analysis of experimental session features identified key phases. High-Frequency Power (HFP) indices from HRV during the patient's speech ("sp2") and recovery ("r") phases emerged as significant predictors across at least 10 of the 11 models tested, spanning both patient- and caregiver-level outcomes and modeling approaches. Patient-level coregulation index during patient speech ("sp2") also showed significance across 10 of the 11 models. Coregulation and coagitation indices at the caregiver level showed similar significance, but these effects across outcomes were mostly confined to the patient-level and did not generalize to their own outcomes. The only exception to this is caregivers' HFP in the recovery ("r") phase, where this predictor was found significant at both levels across 6 of the 11 models tested. In general, HFP and CLO-based gamma features ranked lowest, with some phases (such as baseline "b" and scenario "s" for both patients' and caregivers' HFP) yielding few significant features.

SHAP analysis in best-fitted models for patient-level outcomes revealed two key patterns. First, these models included the full set of coregulation and coagitation indices and nearly all these interrelated indices ranked as highly important within their respective models. Second, the directional influence of these indices frequently shifted depending on experimental phase, despite measuring the same underlying construct. For

Fig. 3. SHAP heatmap of predictor importance from the STITCH task for patients (pt) and caretakers (fm). Rows represent significant predictive models (RF, SVR, LR, RIDGE) and columns are experimental phases. Color indicates directional impact (blue: positive, purple: negative). Gray or blank cells denote non-significant or predictors eliminated by preprocessing, respectively.

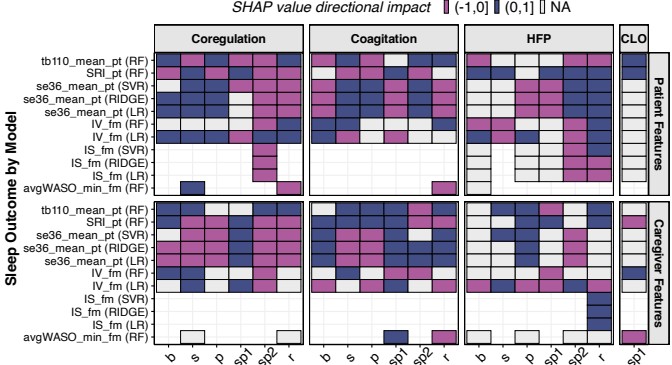

instance, while several patient-level coregulation indices (e.g., baseline "b", scenario "s", prep "p") were positively associated with higher patients' sleep efficiency, this relationship flipped during other phases (e.g., patient speech "sp2" and recovery "r"), becoming negatively associated. Such widespread importance of highly correlated predictors and phase-dependent shifts in directional influence strongly suggests the presence of complex interactions that can be driven by suppressor effects.

To probe the interplay between coregulation and coagitation indices more rigorously, we formally tested for multivariate suppression with respect to the LR model predicting patients' sleep efficiency (se36_mean_pt). We employed Commonality Analysis (CA) using the commonalityCoefficients command from the yhat package in R to quantify suppression effects specifically among coregulation and coagitation predictors. As CA grows exponentially with the number of predictors, we repeatedly sampled random subsets of six indices (out of 24 total) and computed their pairwise commonality coeffi-

Fig. 4. Suppression network of predictors for patients' sleep efficiency. Nodes represent coregulation (coreg) or coagitation (coag) predictors from patients (pt1) and caretakers (fm1) in some experimental phase. Node color denotes community membership as detected by a walktrap algorithm. Edges show average suppression between predictors, with thickness proportional to the mean percentage of total variance such suppression explains and color denoting significance (blue: $p < 0.2$; gray: non-significant).

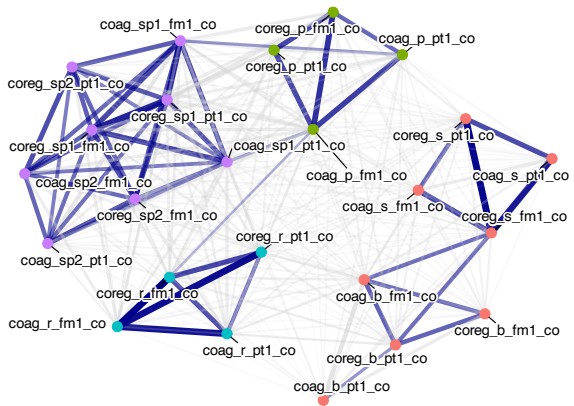

cients across 1,000 resampling iterations to estimate observed negative commonality (i.e., suppression) effects. To assess significance at the edge level, we then performed a permutation test for each predictor pair: within each resampling iteration, predictor labels were shuffled and suppression coefficients recalculated over 1,000 permutations to generate a null distribution of negative commonalities. Figure 4 visualizes the suppression network.

Coregulation and coagitation indices showing theoretically counterintuitive directional effects in the SHAP analysis were found to be embedded within these suppression clusters. For instance, both patient-level and caretaker-level coregulation and coagitation indices during patient speech ("sp2") were situated in the same densely connected cluster together with indices taken during caretaker speech ("sp1"). Such indices taken during recovery ("r") also form their own distinct cluster. This suggests that inconsistent SHAP effects observed among these predictors may result from suppressive interactivity rather than a lack of direct predictive validity.

## IV. DISCUSSION

Our ML framework successfully identified predictive patterns across multiple sleep health outcomes within a novel interdependent dyadic context, with strongest performance in predicting caregivers' interdaily stability and patients' sleep efficiency. It also pinpointed key phases of the experimental session and input modalities that contributed to these predictions, with coagitation and coregulation indices ranking among the most significant predictors. The proposed preprocessing module was crucial in achieving such results, yielding improvements of up to 30.4% compared to baseline methods.

In the pursuit of informing integrative psychosocial correlates of dyadic sleep functioning, our SHAP analysis revealed counterintuitive associations between specific stress indices and sleep health outcomes, effects subsequently attributed via

CA to complex interactions among these highly intercorrelated predictors. A critical finding from CA was that the identified suppressor relationships were predominantly organized temporally, with patient and caretaker features nested together. Such effects confirm that modeling partners' data jointly improves the prediction of self-report sleep health outcomes compared to models that would exclude either partner's data.

The phase-based structure of these interdependent effects offers a bridge to future work. First, it motivates development of ensemble learning methods where individual learners are trained on phase-specific predictor clusters. While individual predictor signs within these learners may still reflect complex interactions, such an approach could help clarify how different phases collectively contribute to sleep outcomes or by identifying dominant phase-specific predictor patterns, without compromising predictive power. Second, the network structure provides an empirical basis for causal inference, informing the specification of path analyses or structural equation models (SEM) to test hypotheses about how dyadic interactions across different phases influence distinct patterns of sleep health.

Strong predictive performance of T1 data also supports using longitudinal forecasting to predict future outcomes. Such an approach is especially valuable for overcoming modeling limitations due to participant attrition at later time points.

## ACKNOWLEDGMENT

The work presented in the paper is supported in part by R01NR016838, PG015434, T32HL007426-45, and NSF-CNS2310807. The authors would like to thank the anonymous reviewers for their constructive feedback on this work.

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
