# OpenReview forum: "An Explainable Machine Learning Framework to Inform Integrative Psychosocial Correlates of Sleep Functioning in Adults with Cancer and their Caregivers"
_IEEE.org/EMBS/BHI/2025/Conference — BHI 2025_

### Official Review · Reviewer_5Qdz · 2025-07-14
**Lack of external validation and clinical comparison limits impact**

**Confidence:** 3
**Clarity Of Writing:** good
**Clinical Significance:** fair
**Methodological Novelty:** good
**Overall Rating:** 6

**Experiments And Results:**

fair

**Questions For The Authors:**

1. How do your ML models compare to existing sleep prediction methods or simple clinical risk scores? Are the computational complexities justified by the performance gains?

**Strengths:**

1. This appears to be the first multi-level ML framework specifically designed to analyze interdependent sleep health within patient-caregiver dyads.
2. The combination of SHAP analysis with Commonality Analysis provides deeper insights into complex predictor interactions, particularly the identification of suppressor effects among dyadic stress regulatory indices.

**Summary Of The Paper:**

This study developed a supervised machine learning framework to predict sleep outcomes in 149 colorectal cancer patient-caregiver dyads (298 individuals). The researchers integrated psychosocial questionnaires, laboratory stress responses, and 20 sleep markers, applying modular preprocessing (PCA, feature selection, data augmentation) with four regression algorithms. Using SHAP analysis and Commonality Analysis, they identified dyadic stress regulatory indices as key predictors with phase-specific suppressor effects.

**Weaknesses:**

1. The best models explain only 17.8-31.2% of variance in sleep outcomes, which may limit clinical utility.
2. The study lacks validation against clinical sleep assessments or comparison with established sleep prediction methods.

---

### Official Review · Reviewer_ombk · 2025-07-16
**ML Approach for Understanding Sleep Health in Cancer Patients**

**Confidence:** 3
**Clarity Of Writing:** good
**Clinical Significance:** great
**Methodological Novelty:** good
**Overall Rating:** 6

**Experiments And Results:**

good

**Questions For The Authors:**

How were the self-report questionnaires validated? Are there concerns regarding the accuracy of self-reported data on sleep and psychosocial factors?
Does the study account for changes over time in sleep health and psychosocial factors? How might the findings differ if assessed longitudinally rather than cross-sectionally?
What measures were taken to minimize biases in data collection and analysis? Are there any potential biases in participant selection or response that could affect the results?
How do the findings translate into practical interventions for improving sleep health among cancer patients and caregivers? Are there specific recommendations based on the results?
Does the study adequately acknowledge its limitations? Are there any critical limitations that the authors may have overlooked?

**Strengths:**

The study uniquely addresses sleep health as a dyadic process, emphasizing the interdependent relationship between cancer patients and their caregivers. This perspective enables a more comprehensive understanding of sleep disturbances and their associated psychosocial factors. The captured dataset for this study is valuable.

**Summary Of The Paper:**

This paper outlines a study focused on sleep disturbances in cancer patients and their caregivers, highlighting the significance of understanding these issues as a dyadic process influenced by daily experiences. The research employs a supervised machine learning framework that integrates multi-modal data from 149 dyads, including psychosocial characteristics and sleep markers. Various preprocessing techniques were applied to enhance model performance, with regression algorithms used to predict sleep outcomes.

**Weaknesses:**

ML approaches, such as Linear Regression and Decision Tree, are explainable. It is not clear how SHAP provided additional value.
PCI reduces data dimension, and as a result, removes the meaning of input features, which is contrary to the explainability focused on in this paper.
The paper's structure is cluttered and makes it difficult for the reader to stay focused.

---

### Official Review · Reviewer_ZeYN · 2025-07-17
**This work presents a machine learning framework (various preprocessing techniques and regression algorithms, alongside Explainable AI (XAI) methods like SHAP and Commonality Analysis) to investigate the interplay of psychosocial factors (multi-modal and multi-level data) and sleep functioning/health in cancer patients and their caregivers. The work highlights the importance of dyadic stress regulatory indices and phase-specific suppressor effects.**

**Confidence:** 5
**Clarity Of Writing:** great
**Clinical Significance:** great
**Methodological Novelty:** good
**Overall Rating:** 7

**Experiments And Results:**

good

**Questions For The Authors:**

Page 3: "... the number of principal components was chosen to explain at least 40% of the cumulative variance.": Could you elaborate on the rationale behind selecting this specific 40% threshold?

**Strengths:**

- The use of multi-modal and multi-level data (psychosocial, cardiovascular, psychological, self-report sleep, actigraphy) allows for a holistic understanding of sleep health.
- The evaluation of various preprocessing techniques (PCA, correlation-based filtering, data augmentation), different regression algorithms (LR, Ridge, RF, SVR), and the usage of Leave-One-Out Cross-Validation (LOOCV) and bootstrapping to quantify uncertainty, given the limited cohort size, is also a strength of the work.
- Clinical relevance

**Summary Of The Paper:**

The paper addresses the sleep disturbance challenge in cancer patients and their caregivers. The methodology involves three optional preprocessing techniques: Principal Component Analysis (PCA) for high dimensionality, correlated feature selection (C) for multicollinearity, and data augmentation (A) for small sample size. Four regression algorithms (Linear Regression, Ridge, Random Forest, and Support Vector Regression) were trained to predict various sleep outcomes. A SHAP analysis on best-fitted models was used to identify key predictors, and Commonality Analysis (CA) was employed to clarify complex interactions among highly intercorrelated multi-level features. The study found that optimal preprocessing combinations significantly improved sleep prediction.

**Weaknesses:**

- The R2 values (31.2% and 17.8%) are not relatively high, suggesting that a large portion of variance remains unexplained (possibly inadequate modelling).
- The work was conducted on a specific cohort of colorectal cancer patients (n=149 dyads), which limits the generalizability of these findings to other experimental conditions.

---

### Official Review · Reviewer_LYoE · 2025-07-18
**A Rich Dataset and Thoughtful Analysis on Stress and Sleep Health in Cancer Patient–Caregiver Dyads**

**Confidence:** 5
**Clarity Of Writing:** good
**Clinical Significance:** great
**Methodological Novelty:** fair
**Overall Rating:** 7
**Final Rating:** 8

**Experiments And Results:**

good

**Questions For The Authors:**

1.	The manuscript states that PCA was applied to questionnaire-derived features with a minimum cumulative variance threshold of 40%. Could the authors justify this choice and provide the variance vs. number of components plot to demonstrate whether an elbow point exists around this threshold? This would help assess whether important variance was discarded.
2.	Since PCA was applied to dummy-encoded questionnaire features (i.e., sparse binary variables), which may not meet the assumptions of PCA, did the authors consider alternative dimensionality reduction methods more appropriate for categorical data, such as Multiple Correspondence Analysis (MCA) or autoencoder-based techniques?
3.	The study integrates multiple data modalities (e.g., binary questionnaire data, continuous physiological indices). How were these feature types normalized or scaled before being input into the machine learning models, and were all features concatenated into a single input vector or processed separately?
4.	Since PCA and feature selection were applied within a leave-one-out cross-validation (LOOCV) framework, were the selected components or features stable across folds? If not, how was interpretability ensured, especially when analyzing feature importance?
5.	The pseudo-class sampling procedure used to balance the training data for regression targets is innovative but may risk data leakage if not applied carefully. Could the authors clarify whether this augmentation was performed entirely within the training folds of LOOCV, and how they ensured that the testing data remained unaffected by synthetic resampling?
6.	The manuscript refers to 20 distinct sleep health outcomes used as target variables, but only a subset are explicitly listed in the results. Could the authors provide a full list of all 20 sleep outcomes, indicating which were derived from actigraphy vs. self-reported sleep diaries, and whether they correspond to patients, caregivers, or both?
7.	The paper discusses modeling interdependence within dyads and includes dyadic stress regulation features, but it remains unclear whether sleep outcome models for patients and caregivers were trained using only individual-level features or full dyadic inputs. Could the authors clarify whether both patient and caregiver features were used when modeling each individual’s outcome?
8.	The SHAP analysis appears to exclude questionnaire-derived PCA components for interpretability reasons. Could the authors clarify whether SHAP was computed on the full model (including PCA features) and filtered afterward, or whether a new model excluding PCA was used? This distinction affects the faithfulness of the reported feature attributions.
9.	Please provide a detailed breakdown of the sample’s demographic characteristics (e.g., gender, race/ethnicity, age,), including data on both patients and caregivers.

**Strengths:**

1.	This work presents a rich, multimodal dataset collected from cancer patient–caregiver dyads, combining questionnaires, physiological stress responses, and actigraphy-based sleep measurements to enable comprehensive modeling of interdependent sleep health.

2.	It systematically compares eight preprocessing configurations, using PCA, correlation filtering, and data augmentation, demonstrating how different strategies affect predictive performance across multiple sleep outcomes.


3.	The study employs rigorous evaluation through leave-one-out cross-validation and bootstrapped confidence intervals, while incorporating SHAP and Commonality Analysis to provide interpretable insights into feature importance and suppressor effects.

4.	It reveals novel, phase-specific interactions between physiological stress indicators and sleep outcomes, offering new evidence for the complex and dynamic nature of stress regulation within caregiving dyads.

**Summary Of The Paper:**

The study introduces a  multi-level machine-learning framework to predict sleep health outcomes in cancer patient–caregiver dyads by integrating questionnaire data, laboratory-induced stress reactivity, and actigraphy-measured sleep. After preprocessing, they tested 640 model configurations (4 preprocessing algorithms × 8 preprocessing setups × 20 sleep outcomes) using LOOCV and bootstrapped R²/RMSE estimates. Twelve models achieved 95% significance, primarily predicting actigraphy-derived rhythm measures. SHAP analysis of physiological stress features revealed phase-dependent effects, and Commonality Analysis uncovered suppressor interactions within stress synchrony features.

**Weaknesses:**

1-The paper emphasizes the dyadic nature of sleep regulation, but it remains unclear whether including the partner’s features meaningfully improves prediction. Could the authors provide ablation results comparing models trained with individual-only features (e.g., patient-only data for patient outcomes) versus dyadic features (patient + caregiver data)? This would directly test the benefit of dyadic modeling over individual baselines.


2- Given that the STITCH task appears to be conducted with both patient and caregiver present in the same setting, could the authors comment on the potential for shared environmental or observational bias? For example, might the co-presence during the speech task amplify physiological synchrony, thereby confounding true individual reactivity with observed dyadic effects? It would be helpful to clarify how the protocol handled physical proximity and whether any adjustments were made to account for this.

3-The manuscript would be stronger with an initial Exploratory Data Analysis to validate its dyadic framework. Specifically, simple analyses like calculating within-dyad correlation of HRV features, plotting synchronized time-series, or using cross-recurrence plots could empirically demonstrate physiological interdependence. This would justify the inclusion of dyadic features and clarify the necessity for suppressor and network analyses.

4-The authors claim to model the interdependent nature of sleep health within patient–caregiver dyads; however, their approach simply feeds features from both individuals into a machine learning model to predict individual outcomes. That’s not a modeling framework that explicitly captures dyadic interdependence, such as actor-partner interdependence models (APIM), joint outcome modeling, or graph-based coupling approaches. Without testing whether partner features improve prediction beyond individual features, or modeling the bidirectional influence between dyad members, the claim of capturing interdependence remains overstated.